# Straightforward synthesis of functionalized γ-Lactams using impure $CO_2$ stream as the carbon source

Yuman Qin [1,2], Robin Cauwenbergh [2], Suman Pradhan [1,2], Rakesh Maiti [1,2], Philippe Franck [2] & Shoubhik Das [1,2] ✉

Direct utilization of $CO_2$ into organic synthesis finds enormous applications to synthesize pharmaceuticals and fine chemicals. However, pure $CO_2$ gas is essential to achieve these transformations, and the purification of $CO_2$ is highly cost and energy intensive. Considering this, we describe a straightforward synthetic route for the synthesis of γ-lactams, a pivotal core structure of bioactive molecules, by using commercially available starting materials (alkenes and amines) and impure $CO_2$ stream (exhaust gas is collected from the car) as the carbon source. This blueprint features a broad scope, excellent functional group compatibility and application to the late-stage transformation of existing pharmaceuticals and natural products to synthesize functionalized γ-lactams. We believe that our strategy will provide direct access to γ-lactams in a very sustainable way and will also enhance the Carbon Capture and Utilization (CCU) strategy.

Direct fixation of $CO_2$ into organic molecules for the synthesis of fineś chemicals and pharmaceuticals is currently an important area[1–7]. The main reason behind this aspect is that $CO_2$ is a nontoxic and abundantly available carbon source. Considering this, a plethora of transformations have recently been achieved despite the challenging activation of $CO_2$ molecule due to its high thermodynamic stability and kinetic inertness[8–17]. In organic synthesis, so far, the major focus has been given towards the formation of novel C-C bonds, C-heteroatom bonds, and others by using this strategy[18,19]. It should be clearly noted that in all of the cases, pure $CO_2$ gas has been used as the carbon source, and to the best of our knowledge, impure $CO_2$ stream such as the direct use of the flue gas collected from industries or exhaust gas collected from a car has rarely been reported. However, if an impure $CO_2$ stream can be used, it certainly avoids the associated cost and energy requirement for the $CO_2$ purification procedure and provides a new direction for the application of the Carbon Capture and Utilization principle[20–25]. Nonetheless, the presence of impurities such as $O_2$, water vapor, CO, NOx, and hydrocarbons in the $CO_2$ stream can be extremely harmful for the catalyst/reactants/intermediates and can be completely detrimental for the product formation.

Parallel to the use of impure $CO_2$ stream, synthesis of γ-lactams is considered as highly important since this scaffold is a ubiquitous active core in diverse natural products and pharmaceuticals (Fig. 1a). Particularly, the functionalized γ-lactams have exhibited a wide variety of biological activities due to their high potency to inhibit enzymes and the growth of the cancer cells[26]. Besides these, the γ-lactam is a pivotal building block for the synthesis of organic molecular entities[27]. Owing to their high importance, γ-lactams are classically constructed intramolecularly by the condensation of amino acid derivatives in the presence of an equivalent amount of coupling reagent, such as N,N'-dicyclohexyl carbodiimide (DCC), acetic anhydride and others (Fig. 1b)[28]. Besides these, the N-alkylation[29–32] and acylation[33,34] of N-protected amides or dioxazolones have also been established as a powerful platform for the synthesis of γ-lactams. However, all these protocols typically required prefunctionalizations of substrates and additionally, faced undesired side reactions. In parallel, an elegant photocatalytic approach where acrylates were used to construct the poly-substituted γ-lactams, has recently been reported. However, the release of a stoichiometric amount

[1]Department of Chemistry, University of Bayreuth, 95447 Bayreuth, Germany. [2]Department of Chemistry, Universiteit Antwerpen, 2020 Antwerpen, Belgium. ✉e-mail: Shoubhik.Das@uni-bayreuth.de

**a. Selected bioactive molecules containing functionalized γ-lactam core**

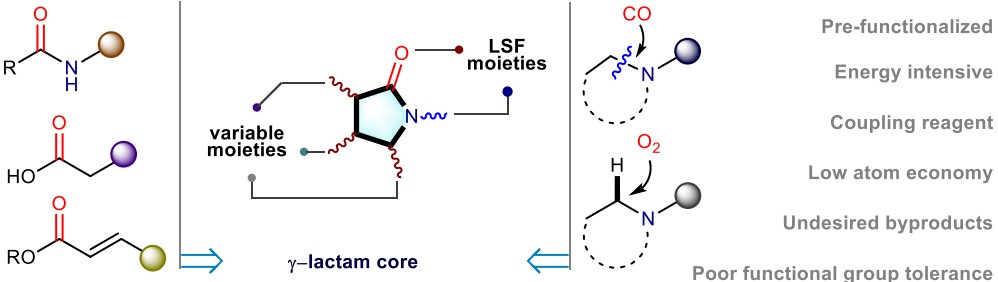

*L-, N- and T-type channel blockers*                    *αvβ3 integrin antagonists*

**b. Existing strategies for the synthesis of γ-lactams**

Pre-functionalized

Energy intensive

Coupling reagent

Low atom economy

Undesired byproducts

Poor functional group tolerance

**c. Existing strategy for the synthesis of γ-lactams using CO₂ as an activator**

Unfunctionalized amines

Exhaust gas as C1 source

HAT

CO₂ as activator

underdeveloped

**d. This work:** photocatalytic synthesis of γ-lactams using exhaust gas as C1 source

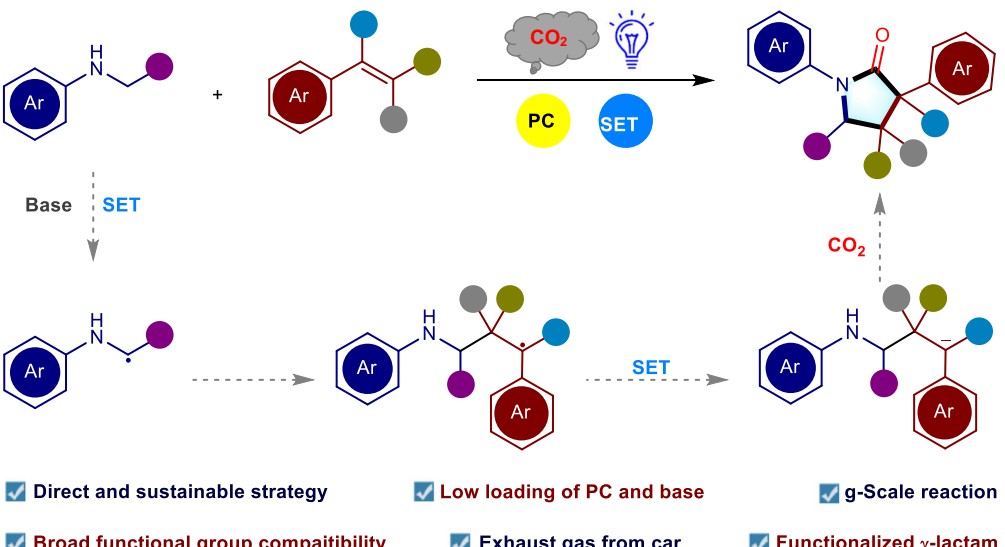

☑ **Direct and sustainable strategy**   ☑ **Low loading of PC and base**   ☑ **g-Scale reaction**

☑ **Broad functional group compatibility**   ☑ **Exhaust gas from car**   ☑ **Functionalized γ-lactam**

**Fig. 1 | Strategies for the construction of γ-lactam core.** LSF late-stage functionalization, PC photocatalyst, SET single electron transfer; **a** selected bioactive molecules containing functionalized γ-lactam core; **b** existing strategies for the synthesis of γ-lactams; **c** existing strategy for the synthesis of γ-lactams using CO₂ as an activator; **d** overview of this work.

of alcohol as a by-product reduced the atom economy of this strategy[35]. Furthermore, the carbonylation reaction using CO has provided an alternative route for the construction of lactam moiety albeit under energy-intensive conditions[36,37]. Moreover, the oxidative transformations of the C-H bond also enabled the conversion of the *N*-heterocyclic motif into corresponding γ-lactam in an atom-economic fashion, however, the oxidizing environment narrowed the substrates scope and limited the

compatibility[38,39]. Therefore, the synthesis of γ-lactams in a straightforward and sustainable fashion from widely available starting materials and carbonyl source should bring significant breakthrough in this domain.

Recently, the photocatalytic transformation of $CO_2$ into value-added products represents a prospective pathway to achieve $CO_2$ transformation/utilization under mild reaction conditions[40–42]. Considering this, the group of Schoenebeck and Rovis jointly reported an elegant synthetic blueprint (Fig. 1c) where $CO_2$ acted as an activator for the respective primary amine, which later reacted with acrylates to produce γ-lactam in the presence of a hydrogen atom transfer catalyst[43]. In this strategy, acrylates were the actual carbonyl source for the synthesis of γ-lactam and the role of $CO_2$ was only to activate the respective amino group. At the final stage, $CO_2$ was released from the corresponding intermediate and thus, no fixation of $CO_2$ molecule occurred to obtain the final product. Furthermore, the research group of Yu has developed a promising approach by utilizing carboxylic acids as bifunctional reagents to synthesize γ-amino acids in a unique $CO_2$-releasing and recapturing manner. One of the examples of this strategy involved the synthesis of a γ-lactam, however, it required three steps: at first, the synthesis of the γ-amino acid occurred, then esterification with $TMSCHN_2$, and finally, acid-mediated deprotection followed by a cyclization reaction took place to produce the corresponding γ-lactam[44]. Followed by this strategy, Sun group has established a γ-amino acid synthesis platform that utilized sodium glycinates as an amino source precursor in the presence of an external $CO_2$ to form γ-amino acid and lactams. However, this strategy also required multiple steps to achieve the γ-lactams and additionally, a relatively longer reaction time (48 h)[45]. Particularly, heating the reaction at 80 °C in the presence of (over) stoichiometric amount of acetic anhydride was essential to form the γ-lactam in this strategy.

Considering all these above-mentioned strategies, we became interested to synthesize γ-lactams from commercially available reagents and using impure $CO_2$ stream as a carbon source. The limited availability of amino acids (Multiple steps synthetic routes developed by the group of Yu[44] and Sun[45]), directed us to utilize amines and that should open the door towards unlimited varieties of γ-lactams in a single step. Particularly, amines are able to generate the corresponding reactive α-amino radicals in a direct fashion under visible-light-mediated photoredox catalysis. However, the direct generation of α-amino radicals is primarily dominated by tertiary amines[46–50], and limits the substrates scope as well as diversification via further modifications. Furthermore, tertiary amines will be problematic for the cyclisation step in the synthesis of γ-lactam. In contrast, secondary amines are considered as an ideal amino source due to their tunable -NH group and their wide availability. Certainly, this approach will circumvent the pre-installation of the functional group (such as in case of amino acids, amino esters, α-silylated derivatives, and imines)[49,51–54]. Inspired by this information, we anticipated that a three-component reaction among a carbonyl source ($CO_2$ from the impure $CO_2$ stream), an amino source (secondary amine), and a carbon backbone (alkene) could readily build the γ-lactam core unit under the mild reaction conditions of photoredox catalysis (Fig. 1d). In fact, unprotected secondary amines are able to generate the reactive α-amino radicals and should serve as an appropriate amino source to assemble the γ-lactam ring[55,56]. Later, the presence of an easily accessible alkene will act as the carbon backbone and should facilitate the construction of C-C bond by intercepting the α-amino radical (generated from the secondary amines) and will trigger the generation of the corresponding carbanion via further single electron transfer (SET) process. The carbanion can easily attack the electrophilic carbon center of $CO_2$, will lead to the carboxylation reactions and finally, an intramolecular nucleophilic attack should generate the functionalized γ-lactam.

## Results and discussion

### Reaction condition optimization

At the beginning of our investigation, we chose N-ethyl aniline (**1a**) to react with 1,1-diphenylethylene (**2a**) and $CO_2$ under atmospheric pressure under the irradiation of visible-light. After systematic optimization of the reaction parameters (for detailed information, see Supplementary Table 1–4), we were delighted to find that, in the presence of 0.2 mol% of $Ir(ppy)_2(dtbbpy)PF_6$ and 20 mol% of $KHCO_3$, in N,N-dimethylformamide (DMF) under a 40 W blue Kessil lamp ($\lambda = 456$ nm) at ambient temperature, the desired product was obtained in 98% yield (Table 1, entry 1). We rationalized that the photocatalyst (PC), solvent and the base might play the central role for the generation of the reactive α-amino radical species from **1a**[50,57]. For this reason, we set out to investigate a range of PCs (Table 1, entries 2–4) and unfortunately, none of them reacted similar to the model PC. The underlying rationale behind this disparity can be attributed to the close proximity of the redox potential of $[Ir(ppy)_2dtbbpy]PF_6$ ($Ir(III)^*/Ir(II) = +0.66$ V vs SCE, $Ir(III)^*/Ir(II) = -1.51$ V vs SCE)[47] to both the oxidizing potential of the secondary amine (+0.81 V vs SCE for N-methylaniline)[35] and the reducing potential of the benzylic radical (−1.61 V vs SCE for phenylethyl radical)[40]. This intriguing fact presents an opportunity to successfully complete the catalytic cycle, given the appropriate optimization of reaction conditions. While the inefficacy of *fac*-$Ir(ppy)_3$ in promoting the reaction lies in the fact that the excited state of *fac*-$Ir(ppy)_3$ ($Ir(III)^*/Ir(II) = +0.31$ V vs SCE)[47] is incapable of oxidizing secondary amine to generate the α-amino radical. This strategy clearly brings an alternative pathway developed by the Xi group with tertiary amines in the presence of a 4CzIPN photocatalyst (Table 1, entry 4)[46]. Since the redox property of 4CzIPN is quite limited, it will be problematic for the activation of the secondary amines and reduction of generated benzylic radical in this protocol and therefore, secondary amines did not exhibit any reactivity in their protocol.

Different solvents such as dimethyl sulfoxide and dimethylacetamide were also evaluated (for detailed information, see Supplementary Table 2), and turned out to be slightly less efficient (Table 1, entries 5–6). This phenomenon could be attributed to the high solubility of $CO_2$ in DMF. Additionally, further optimizations of the quantity of $KHCO_3$ (for detailed information, see Supplementary Table 3) and reaction temperature (for detailed information, see Supplementary Table 4) were also performed. Furthermore, control experiments were conducted to determine the role of each reaction parameter (Table 1, entries 7–10), and revealed that no product was formed in the absence of the PC, base, $CO_2$, or visible light. Gratifyingly, the gram-scale conversion proceeded with an excellent yield (92%) by using this photoredox strategy, which clearly demonstrated the practicality and scalability of our recetion strategy (Table 1, entry 11). Most interestingly, the use of exhaust gas (containing a mixture of $CO_2$: 10.8414%, $N_2$: 61.0585%, CO: 0.0054%, $O_2$: 2.4532%, $CH_4$: 0.0061% and others: 25.6354%) from a car instead of pure $CO_2$ in this reaction yielded the desired product (**3a**) with a surprisingly high yield (85%, Table 1, entry 12), which exhibited the robustness as well as practical deployment of our protocol to synthesize this scaffold.

### The substrate scope in anilines and alkenes

After the optimization of the reaction conditions, we became interested to demonstrate the generality of this strategy by varying different amine derivatives in this reaction (Fig. 2). To our observation, different N-alkyl anilines (**3a** – **3c**), alkyl chain substituted with secondary alkyl group (**3d**), bulky tertiary alkyl group (**3e**) and phenyl group (**3f**) afforded a series of γ-lactams in good to excellent yields (65–98%). Alongside, several N-alkyl aniline derivatives with electron withdrawing groups (EWGs) and electron donating groups (EDGs) in the aliphatic chain, including -OTBDMS (**3g**), -OMe (**3h**), -(OEt)$_2$ (**3i**), -CN (**3j**) also provided the corresponding γ-lactams with good to excellent yields (60–93%). Notably, this conversion was also amenable

**Table 1 | Optimization studies for the synthesis of γ-lactam**

$CO_2$ (balloon)
[Ir(ppy)$_2$(dtbbpy)]PF$_6$ (0.2 mol%)
KHCO$_3$ (20 mol%)
DMF (0.1 M)
Kessil lamp (λ = 456 nm)
20 h, r.t.

| Entry[a] | Variation from standard condition | Yield[c] (%) |
|---|---|---|
| 1 | none | 98 (90)[d] |
| 2 | Ir(dF(CF$_3$)ppy)$_2$(dtbbpy)PF$_6$ | 45 |
| 3 | fac-Ir(ppy)$_3$ | 0 |
| 4 | 4CzIPN | 0 |
| 5 | DMA | 78 |
| 6 | DMSO | 70 |
| 7 | no PC | 0 |
| 8 | no K$_2$CO$_3$ | 0 |
| 9 | under dark | 0 |
| 10 | under N$_2$ | 0 |
| 11[b] | 10 mmol scale | 92 |
| 12 | exhaust gas | 85 |

[a]Reaction conditions: **1a** (0.24 mmol, 1.2 equiv.), **2a** (0.2 mmol, 1.0 equiv.), PC (0.2 mol%), KHCO$_3$ (20 mol%), DMF (2 mL), 40 W Kessil lamp (λ = 456 nm), CO$_2$ (balloon), room temperature (r.t.), 20 h.
[b]Reaction conditions: **1a** (12 mmol, 1.2 equiv.), **2a** (10 mmol, 1.0 equiv.), PC (0.2 mol%), KHCO3 (20 mol%), DMF (50 mL), 40 W Kessil lamp (λ = 456 nm), CO$_2$ (balloon), room temperature (r.t.), 72 h.
[c]The yield was determined by 1H NMR using 1,3,5-trimethoxybenzene as the internal standard.
[d]Isolated yield.

to substrates with alkene (**3k**) as well as protected piperidine (**3l**) to provide the desired products in excellent yields (93% and 70%). Additionally, anilines bearing EDGs at the para position (**3m** – **3p**) afforded the products in good to excellent yields (63–93%), halide-substituted aniline derivatives (**3q** – **3t**) and EWGs (**3u**) also delivered the desired products in good to excellent yields (63–91%). Moreover, the change in substituents pattern i.e., meta-substituted aniline with EDGs (**3v**, **3w**) displayed excellent reactivity under this reaction conditions (89%, 84%). Interestingly, even the polyhalides (**3x**) provided an excellent yield (73%). Delightfully, the heterocyclic amines containing γ-lactams (**3y**, **3z**) can also be synthesized in excellent yields (95%, 84%). Consistent with the reactivity of carbon radical, the N-methylaniline with primary reaction site was also able to furnish the product (**3aa**) in a slightly compromised yield (67%). Remarkably, the substrate bearing a carboxylic acid group (**3ab**, 40%) was able to the protonate the in situ generated carbanion intermediates, was also tolerated in this system, indicating the high efficiency and compatibility of this strategy. Surprisingly, the employment of heteroaryl amines led to the construction of seven-membered ε-lactam (**3ac**, 75%), highlighting the diverseness and synthetic utility of this methodology.

Furthermore, the potential utility of this synthetic strategy was also exemplified by varying the structure of alkenes (Fig. 2). In fact, 1,1-diarylethylenes bearing EWG (**4a**, 75%) at the para position exhibited good reactivity in this system. On the other hand, in terms of regioselectivity, 1,1-diarylethenes with different substituents, two regioisomeric products were obtained in good to excellent yield. The substrates with -EDGs at the ortho (**4b**) or para (**4c**, **4d**) position, delivered the corresponding products in good to excellent yields (65–85%). Furthermore, an internal vinyl arene underwent this conversion to provide the targeted product in good yield (**4e**, 68%). Consistent with the electronic arguments, the aryl alkenes bearing both strong and weak EWGs (**4f** – **4k**) were excellent partners and provided γ-lactams in good to excellent yields (61–89%). Remarkably, the electron-neutral 2-vinylnaphthalene also afforded the desired product in an acceptable yield (**4l**, 40%). Delightfully, the yield was

driven high by the methyl substituted external 2-vinylnaphthalene (**4m**, 83%), which could be due to the higher stability of the secondary carbon radical. The electron-neutral substrate was also compatible with the reaction conditions and exhibited the targeted γ-lactam in moderate yield (**4n**, 33%). However, there was a significant improvement when an EWG was grafted onto the substrate (**4o**, 91%), specially, the biologically relevant vinyl pyridine (**4p**, 95%) was well tolerated.

**Late-Stage Transformations (LST)**

Having demonstrated the scope of this protocol, late-stage transformations (LST) of biologically active molecules were carried out to introduce the γ-lactam core in the pharmaceutically active compounds. Since γ-lactam core has the potential to improve the antibiotic, antitumor, and antidepressant properties in drug molecules[26,58], we rationalized that our straightforward synthetic strategy should open a new avenue to enhance the drug discovery via LST. As shown in Fig. 3, the construction of γ-lactam scaffold in cholesterol derivative (**5a**, 49%) opened a new window for the delivery of anticancer, antimicrobial, antioxidant drugs and cholesterol-based liquid crystals and gelatos[59,60]. The (+)-dehydroabietylamine was also effectively functionalized to **5b** in 65% yield which might become an interesting candidate for fragrance and triple-negative breast cancer treatment[61]. Furthermore, a γ-lactam center was successfully installed in varieties of natural products such as oleylamine (**5c**), (-)-cis- myrtanylamine (**5d**), tocopherol (**5e**), and nerol (**5f**) in good to excellent yields (43 – 85%).

Gratifyingly, a variety of bioactive alkenes such as L-menthol (**5g**) and fenofibrate (**5h**) were also converted to the corresponding γ-lactam derivatives with good to excellent yields (65% and 86%). Furthermore, when flurbiprofen and dexibuprofen derivatives were subjected to the reaction conditions, the corresponding γ-lactam (**5i**, **5j**) were obtained in excellent yield (71% and 75%), paving a new opportunity to synthesize profen analogs. Moreover, this strategy was further extended to introduce a γ-lactam core between complex molecules with different biotically features to provide the corresponding conjugated compounds (**5k** – **5m**, 40–53%). Expediently, the LST of various complex molecules underwent excellently when the

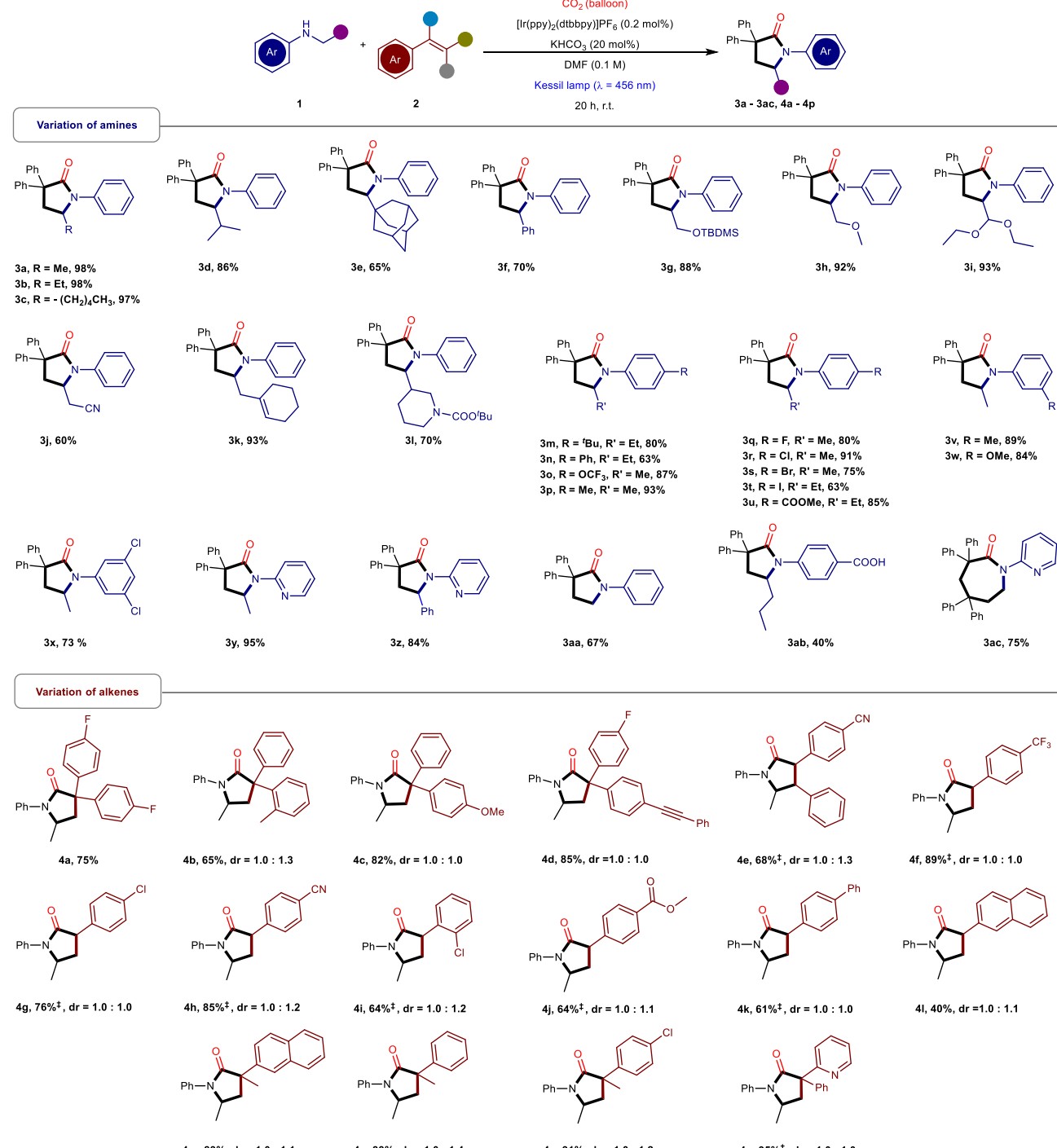

**Fig. 2 | The substrate scope of amines and alkenes.** Reaction condition: as mentioned in Table 1, Entry 1; ‡ $K_3PO_4$ was used instead of $KHCO_3$.

exhaust gas (containing a mixture of $CO_2$, $N_2$, $H_2O$, CO, Nox, and $O_2$) was considered as the carbon source instead of pure $CO_2$ (**5bb** – **5db, 5gb, 5ib, 5jb**; 49–71%). This clearly demonstrated that the rapid diversification and construction of functionalized γ-lactams with a large biological spectrum and therapeutic applications could be afforded from the exhaust gas. We strongly believe that this new blueprint will certainly open a new era for the synthesis of drug molecules via the valorization of waste gases.

## Mechanistic investigation

To shed light on the reaction pathway, a range of mechanistic experiments were carried out. At first, radical control experiments

were performed, which inhibited the formation of the targeted product in the presence of different equivalents of 2,2,6,6-Tetra-methylpiperidinooxy (TEMPO), indicating the involvement of a radical process (Supplementary Table 5). Similarly, the employment of $CuCl_2$ as a quencher identified the presence of a SET process (Supplementary Table 5, entry 3). In order to understand the initial step of this transformation, Stern-Volmer fluorescence quenching experiments were also carried out (for details, see Supplementary Fig. 6). In fact, fluorescence intensity decreased with the increase of N-ethyl aniline concentration, however, no change was observed with 1,1-diphenylethylene, which clearly revealed that the PC after the excitation, was quenched by the N-ethyl aniline. We assumed that the

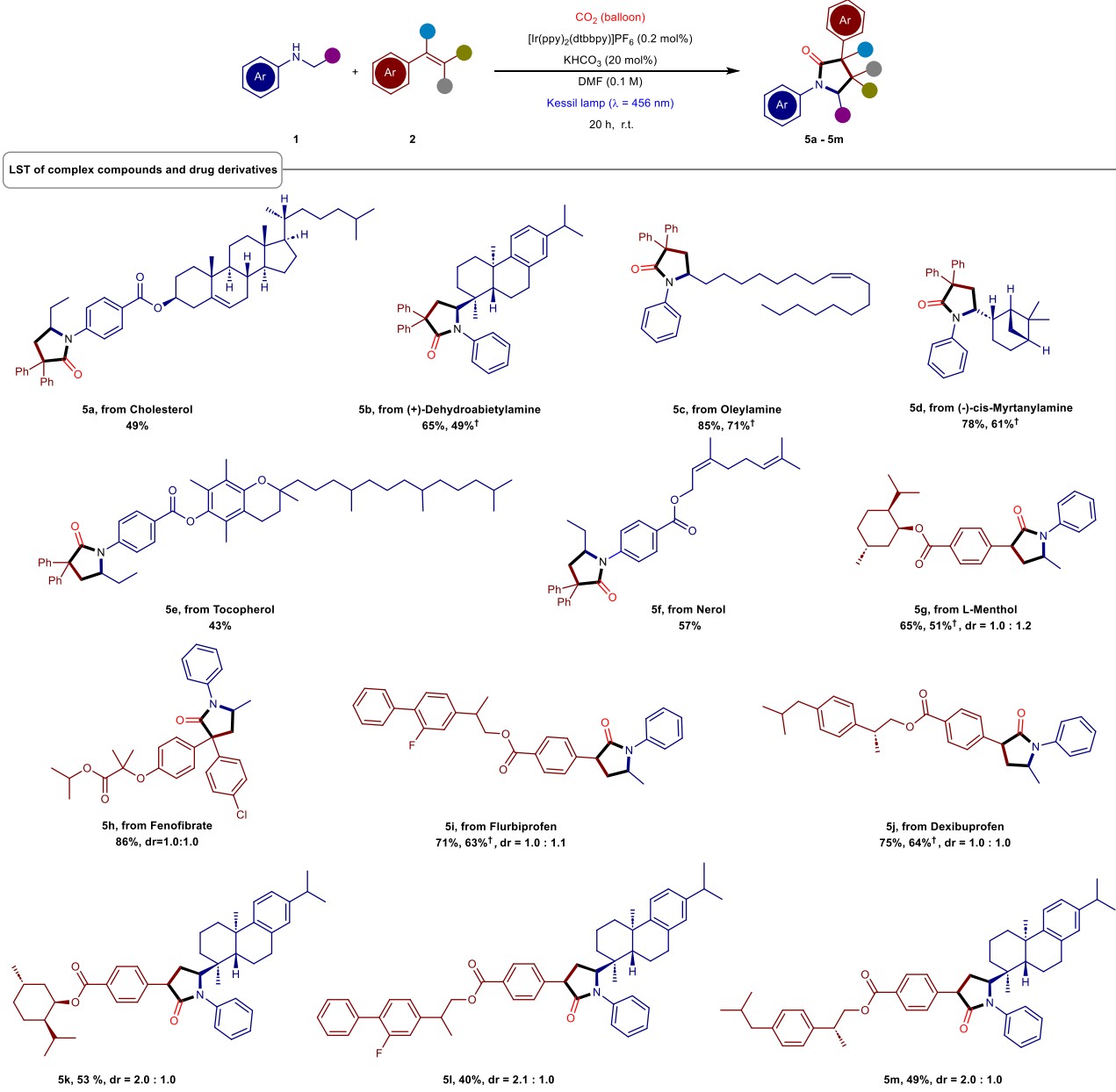

**Fig. 3 | Synthesis of functionalized γ-lactams using the exhaust gas.** Reaction conditions: as mentioned in Table 1, Entry 1; † Using exhaust gas from car.

excited state of the PC acted as an oxidant to accept an electron from N-ethyl aniline, followed by losing a proton in the presence of a base, generated the corresponding α-amino radical[34,35]. In fact, the generation of the corresponding dimer **7a** was also observed in certain reaction conditions which was the direct evidence for the presence of the α-amino radical (Fig. 4a). Additional experiments were also performed to gain insight into other intermediates formed in this process. The isotope labeling experiments with $D_2O$ yielded up to 90% of deuterium incorporation, implying that γ-amino benzylic anion species could be the active intermediate (Fig. 4b). This result was in consistent with the formation of γ-amino alcohol in moderate yield when heptanal was added as an electrophile (Fig. 4c). The defluorinated alkylation of trifluoromethyl alkene experiment further supported the presence of γ-amino benzylic anion intermediate (Fig. 4d). Furthermore, $^{13}CO_2$ labeling experiments clearly revealed that $CO_2$ was the unique carbon source of the carbonyl moiety of the corresponding γ-lactam (Fig. 4e). Moreover, light on/off experiments indicated that the formation of the product was completely inhibited

in the absence of light but was able to restart when the light was switched on (For details, see Supplementary Figs. 9, 10).

Based on all these mechanistic investigations, the proposed mechanism has been outlined in Fig. 5. Initially, the [Ir^III] photocatalyst was excited under the irradiation of visible light. The excited [Ir^III]* complex acted as an oxidant to accept an electron from **1a** to generate the reduced [Ir^II] together with the formation of species **A**. Subsequently, **A** lost a proton with the assistance of $KHCO_3$ to provide the radical **B**, which subsequently reacted with **2a** to afford the radical species **C**. This radical species underwent a SET reduction process with [Ir^II] to produce γ-amino benzylic anion species **D**. In the presence of $CO_2$, species **D** attacked the electrophilic carbon center of $CO_2$ to generate the corresponding carboxylate **E**. Followed by this, an intramolecular nucleophilic attack on the carboxylic acid group occurred to produce the corresponding γ-lactam product.

In summary, we have developed a photoredox strategy for the synthesis of γ-lactams using easily available amine, alkene, and $CO_2$. Importantly, this photoredox strategy exhibited a broad substrate

**a. Intercepting the α-amino radical**

**b. The isotope labelling experiment with D₂O**

**c. Intercepting the carbanion intermediate with aldehyde**

**d. Defluorinative alkylation of α-(Trifluoromethyl)-styrene**

**e. ¹³CO₂ labelling experiment**

**Fig. 4 | Mechanistic studies. a** intercepting the α-amino radical; **b** the isotope labeling experiment with D₂O; **c** intercepting the carbanion intermediate with aldehyde; **d** defluorinative alkylation of α-(Trifluoromethyl)-styrene; **e** ¹³CO₂ labeling experiment.

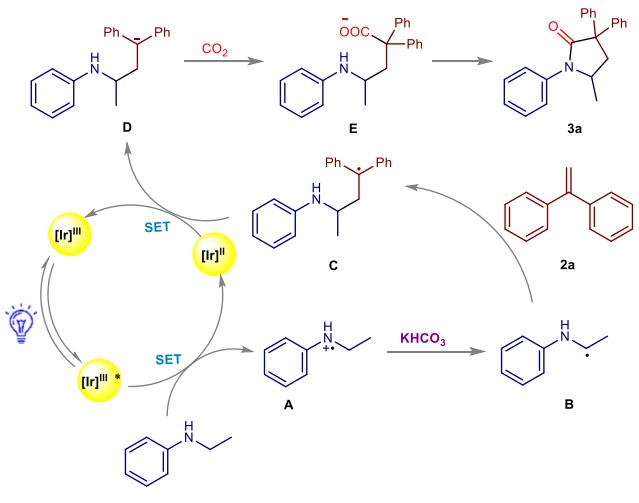

**Fig. 5 | Proposed mechanism for the synthesis of γ-lactams.** [Ir]: Ir(ppy)₂(dtbbpy) PF₆; SET: single electron transfer; The mechanism was proposed with model substrates **1a** and **2a**.

scope and the application of this system has been diversified by applying onto complex molecules which highlighted the utility of this strategy for the construction of a library of bioactive analogs.

Moreover, this system can be readily scaled-up, which indicated the strong potential in industrial production. Additionally, detailed mechanistic studies revealed the mechanism of this transformation. However, the central advantage of this strategy is that the exhaust gas can replace pure $CO_2$ for the synthesis of pharmaceutically active compounds which certainly will avoid the all the energy and cost prohibitive $CO_2$ purification procedure.

## Methods

A dry 10 mL vial equipped with a stirring bar was charged with the 1,1-diphenylethylene (0.2 mmol), N-ethylaniline (0.24 mmol), [Ir(ppy)₂(dtbbpy)]PF₆ (0.2 mol%), and KHCO₃ (20 mol%). Subsequently, dimethylformamide (DMF) with a concentration of 0.1 M was introduced as the solvent. The vial was sealed with a rubber plunger, and the reaction mixture was subjected to a continuous stream of carbon dioxide ($CO_2$) for a duration of 10 min. Following complete saturation of the reaction mixture with $CO_2$, the vial was placed approximately 3 cm away from a Kessil lamp (λ = 456 nm), and the mixture was stirred under the irradiation with a fan for a period of 20 h. Subsequent to the irradiation, the mixture was quenched by dilution with a 0.1 M HCl solution (2.0 mL) and ethyl acetate (EtOAc, 2.0 mL). 1,3,5-Trimethoxybenzene was added, and the resulting layers were separated. The aqueous layer was further subjected to extraction with ethyl acetate (2.0 mL x 3), and the combined organic layers were concentrated under reduced pressure using a 40 °C water bath. The yield was analyzed by ¹H NMR spectroscopy.

## Data availability

The data supporting the results of this work are included in this paper or in the Supplementary Information and are also available upon request from the corresponding author.

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

## Acknowledgements
Y.Q. acknowledge funding from the Chinese Scholarship Council (PhD fellowship to Y.Q., grant no: 202006870012). R.C. acknowledge funding from the FWO (PhD fellowship to R.C.). S.D. thank Francqui foundation (lecturer award to S.D.) Odysseus grant and FWO research project grant. We thank Tong Zhang and H. Y. Vincent Ching for their kind assistance in Stern-Volmer fluorescence quenching experiments.

## Author contributions
S.D. and Y.Q. conceived and designed the experiments. Y.Q. and R.C. performed the substrate scope experiments. Y.Q., S.P. and R.M. carried out the mechanistic studies. P.F. collected the exhaust gas. All authors critically reviewed the manuscript and approved the final version.

## Funding

## Competing interests
The authors declare no competing interests.
