## [Peer Review File · Nature Communications]

Straightforward Synthesis of Functionalized γ -Lactams using Impure CO₂ Stream as the Carbon SourceReviewers' Comments:

Reviewer #1:

Remarks to the Author:

This manuscript describes the three-component coupling reactions of (exhaust) CO₂, 2° aromatic amines (aniline derivatives), and aryl-(di)substituted olefins using an Ir photosensitizer under photo-irradiation. A novel and noticeable aspect (as a very rare example) of the present coupling reaction to form γ -lactams is no stoichiometric amounts of salt waste generated after the reaction since KHCO₃ was used in a catalytic amount (20 mol %). Unfortunately, in many other preceding C–C bond formation reactions to produce carboxylic acids (CAs) from CO₂ for comparison, carboxylate salts are majorly produced since (over)stoichiometric base additives are frequently needed, although CAs, rather than the corresponding carboxylates, are the desired end products. To avoid this drawback of salt formation, the authors ingeniously incorporate a process of intramolecular cyclization to form lactam rings. A vault of Ar-(di)substituted olefins was only tested since the mechanism is seemingly involving benzylic radical intermediates, which are rather long-living species that could thermally accept one electron transferred from the one-electron reduced species (OERS) of the Ir(III) photosensitizer (Ir(II) species) to give the carbanion that could rapidly trap CO₂ (in a two-electron reduction manner which is much more favorable) or a proton. Why only aniline derivatives could be used or were examined is unclear, and this limitation ruins the synthetic value of the present results since the nitrogen atom of the 3° amide of lactam is always dangling an N-aryl group, which cannot be easily deprotected. However, from a viewpoint of structure diversity-oriented synthesis for finding a new drug, a variety of highly functionalized compounds was able to be transformed into lactams using the present photocatalytic method, which was demonstrated to be highly viable for the late-stage functionalization of complex molecules potentially of pharmaceutical importance. Overall, this reviewer could recommend the publication of this manuscript in Nature Communication after the following parts are reconsidered by the authors.

- 1) Why Ir(ppy)₃ and 4CzIPN were unable to promote the coupling reaction? Oxidation of 2° amines did not proceed? The thermodynamic (redox potential) relationships among reaction partners should at least be provided to convince a broader readership. They are among the best-reducing agents when they are photo-excited (e.g., ACS Catal. 2022, 12, 15400). König has already demonstrated the effectiveness of the OERS of 4CzIPN generated when used with catalytic thiol (ref. 45) for reducing a benzyl radical to the benzyl anion.
- 2) The accurate gas composition (ratio) of different gases in CO₂ should be measured and provided.
- 3) The authors are requested to use an aniline derivative whose N-Ar group is easily deprotected. For example, N-4-methoxyphenyl or its derivatives and check the deprotection to have an NH group. Can compound 3o undergo N-Ar-deprotection?
- 4) The role of catalytic KHCO₃ is unclear. How does it work in the catalytic cycle? Since a 2 equiv and 0.2 equiv of KHCO₃ did not change or affect the reactivity (SI, page S7), the bicarbonate seems not simply working as a deprotonating agent. Could you further reduce the amount to 0.1 equiv or even lower?
- 5) Refs. 64 and 73 are duplicated and ref. 52 has the wrong authors' list. The authors should check all the references again as carefully as possible.
- 6) I know it is very hard to answer but one of the most important questions in my view is: when we think about CO₂ immobilization into an organic framework via C–C bond-forming strategy, we frequently need to fabricate kinetically a carbanion from a carbon-centered radical. Why the Ir photosensitizer used by the authors is doable kinetically for this in the present system? Thermodynamic (redox) relationships of each component involved in the catalytic cycle should at least be shown in the text.
- 7) Compounds 5a and 5e: The structures should be further unambiguously ascertained by the elementary analysis after their isolation since HRMS-ESI analysis did not work for them.

Reviewer #2:

Remarks to the Author:

This paper presents interesting topic and nice results on straightforward synthetic protocol of γ -lactams, a pivotal core structure of bioactive molecules, by using commercially available starting materials (alkenes and amines) and impure CO₂ stream (exhaust gas collected from the car) as the carbon source. The central advantage of this strategy is that the exhaust gas can replace pure CO₂ for the synthesis of pharmaceutically active compounds. As a consequence, the paper is striking enough and well-organized to warrant publication. I do recommend it acceptance for publication after minor revisions.

Sure, impure CO₂ directly from flue gas or other sources has great practical application due to avoidance of the associated cost and energy requirement for the CO₂ purification procedure and provides a new direction for the application of the Carbon Capture and Utilization (CCU) principle. However, it also faces challenging, because the presence of impurities such as O₂, water vapor, CO, NO_x, and hydrocarbons in the CO₂ stream can be extremely harmful for the catalyst/reactants/intermediates and can be completely detrimental for the product formation. In this context, designing the reactions and catalyst systems would be crucial. Please add comments and discussion on how to deal with these issues to circumvent impurity problems in this work.

Additional discussion at the beginning of the Introduction on carbon capture and utilisation (CCU) (integration of CO₂ capture and subsequent conversion) to organic compounds such as HCCOH, MeOH and organic molecules can be traced back to 2011 (e.g. CO₂ capture and activation and its subsequent conversion, *Energy Environ. Sci.*, 2011, 4, 3971; *Angew. Chem. Int. Ed.*, 2012, 51, 11306), and recently-published account (see, *Green Chem.*, 2016, 18, 5831; *Acc. Chem. Res.*, 2019, 52, 2892-2903, and references are therein) is also suggested.

On the other hand, identification or trapping the radical species C, γ -amino benzylic anion species D is suggested to be done further, by using EPR or in situ spectroscopic techniques such as operando DRIFTS, or DFT study.

Reviewer #1

1) Why Ir(ppy)₃ and 4CzIPN were unable to promote the coupling reaction? Oxidation of 2°amines did not proceed? The thermodynamic (redox potential) relationships among reaction partners should at least be provided to convince a broader readership. They are among the best-reducing agents when they are photo-excited (e.g., ACS Catal. 2022, 12, 15400). König has already demonstrated the effectiveness of the OERS of 4CzIPN generated when used with catalytic thiol (ref. 45) for reducing a benzyl radical to the benzyl anion.

Response: Thank you very much for this constructive suggestion. **Table 1** represents the redox potentials of fac-Ir(ppy)₃, 4CzIPN, [Ir(ppy)₂dtbbpy]PF₆, N-methylaniline, and the benzylic radical. The underlying reason for the inefficiency of fac-Ir(ppy)₃ due to the fact that the excited state of fac-Ir(ppy)₃ (Ir(III)/Ir(II) = + 0.31V) is incapable of oxidizing N-methylaniline (+ 0.81 V) to generate the α-amino radical. While 4CzIPN (PC*/PC⁻ = + 1.35V) can indeed initiate the first step by oxidizing N-methylaniline, it falls short in reducing the benzylic radical to benzyl anion. It is worth noting that the formation of the diamine compound (**7a**, α-amino radical homo coupling) was indeed observed when using 4CzIPN as the PC. It is also true that 4CzIPN has demonstrated its capability to affect the reduction of benzyl radical to the corresponding benzyl anion in numerous well-established approaches. However, the expected effective role of 4CzIPN was not realized in our system. As the referee suggested, we have added the redox potential of every reaction partner in the manuscript and the relevant articles have been cited.

Table 1. The redox potential of different PC and starting material^a.

PC	E _{1/2} (PC*/PC ⁻)	E _{1/2} (PC/PC ⁻)	E _{1/2} (ArNHR/ArNHR [•]) ^b	E _{1/2} (Bn [•] /Bn ⁻) ^c
fac-Ir(ppy) ₃	+0.31V	-2.19V	+0.81V	-1.60V
4CzIPN	+1.35V	-1.21V		
[Ir(ppy) ₂ dtbbpy]PF ₆	+0.66V	-1.50V		

a: *E*_{1/2} vs SCE; b: *E*_{1/2} vs SCE for the N-methylaniline; c: *E*_{1/2} vs SCE for the phenylethyl radical.

2) The accurate gas composition (ratio) of different gases in CO₂ should be measured and provided.

Response: Thank you very much for the referee's valuable suggestions. The process of exhaust gas collection is illustrated in Fig. 1. Upon successful collection, the gas composition was analyzed using Gas Chromatography (GC) techniques. The constituents present in the exhaust gas are depicted in Figure 2, with the respective proportions of each gas detailed in Table 2. These specific data points have been included in the manuscript and highlighted with yellow color.

Fig. 1 Collecting procedure of exhaust gas.

Fig. 2 GC spectrum of exhaust gas.

Table 2. The composition of exhaust gas.

Entry	Components	Percentage (%)
1	CO ₂	10.8414
2	CO	0.0054
3	O ₂	2.4532
4	N ₂	61.0585
5	CH ₄	0.0061
6	Others	25.6354

3) The authors are requested to use an aniline derivative whose N-Ar group is easily deprotected. For example, N-4-methoxyphenyl or its derivatives and check the deprotection to have an NH group. Can compound 3o undergo N-Ar-deprotection?

Response: Thanks for this suggestion. Initially, we devised the reaction employing aniline derivatives featuring Ar groups. The diverse array of substituent groups available for Ar imparted broad applicability to this strategy, potentially leading to bioactive properties. As the referee suggested, the N-Ar deprotection step offers an avenue to access NH lactam, a valuable core structure within medicinal molecules. The N-dearylation process has been developed very well. Such as: Treating the γ -lactam

products with phenyliodinediacetate (PIDA) or phenyliodine bis(trifluoroacetate) (PIFA) in a mixture of H₂O and MeCN for 5 min removed the arylmoieties to afford NH-lactam (Rong, H., Cheng, Y., Liu, F., Ren, S., & Qu, J. J. *Org. Chem.* **82**, 532 (2017)).

4) The role of catalytic KHCO₃ is unclear. How does it work in the catalytic cycle? Since a 2 equiv and 0.2 equiv of KHCO₃ did not change or affect the reactivity (SI, page S7), the bicarbonate seems not simply working as a deprotonating agent. Could you further reduce the amount to 0.1 equiv or even lower?

Response: Thanks for this suggestion. We propose that this system should operate within an alkaline environment, which is helpful for the deprotonation step and cyclization step. The inclusion of KHCO₃ serves to regulate the pH value within this system. Notably, the inherent basic nature of the diverse aniline substrates also contributes to pH adjustment. When the pH level aligns with the conditions conducive to lactam formation, the additional base does not exert any significant influence on the yield. This rationale elucidates the consistent reactivity observed even with the utilization of 2 equiv or 0.2 equiv of KHCO₃. Conversely, employing a lower quantity of KHCO₃, such as 0.1 equiv or less, fails to attain the requisite pH range for optimal performance.

5) Refs. 64 and 73 are duplicated and ref. 52 has the wrong authors' list. The authors should check all the references again as carefully as possible.

Response: Thanks very much for this careful inspection. The authors' list in ref. 52 has been corrected and the ref 73 has been deleted. All the references have been rechecked and renumbered carefully.

6) I know it is very hard to answer but one of the most important questions in my view is: when we think about CO₂ immobilization into an organic framework via C-C bond-forming strategy, we frequently need to fabricate kinetically a carbanion from a carbon-centered radical. Why the Ir photosensitizer used by the authors is doable kinetically for this in the present system? Thermodynamic (redox) relationships of each component involved in the catalytic cycle should at least be shown in the text.

Response: We have incorporated the redox relationships of each constituent into the manuscript. **Table 1** shows the proximity of the redox potential of [Ir(ppy)₂dtbbpy]PF₆ (Ir(III)^{*}/Ir(II) = + 0.66V, Ir(III)/Ir(II) = - 1.51V) to both the oxidizing potential of N-methylaniline (+ 0.81 V) and the reducing potential of the benzylic radical (- 1.61V). This fact provides a chance to complete the catalytic cycle, contingent upon the optimization of reaction conditions. Furthermore, the outcome stemming from the Stern-Volmer experiment, and the comprehensive mechanistic investigations offer compelling validation for the kinetic feasibility of [Ir(ppy)₂dtbbpy]PF₆ within our system.

7) Compounds 5a and 5e: The structures should be further unambiguously ascertained by the elementary analysis after their isolation since HRMS-ESI analysis did not work for them.

Response: Thank you very much for the suggestion. The characterization of bioactive complex molecules often presents challenges related to protonation and stability. We have isolated the products and got pure NMR spectrum (in the supporting information). Through a meticulous comparison and analysis of the spectra, a congruence is observed between the obtained signals and the anticipated structural features. Notably, the resonance at 176.10 ppm (5a, Fig. 3) and 176.08 ppm (5e, Fig. 4) unequivocally corresponds to the amide carbon (C=O in lactam), as explicitly evident in the ¹³C NMR spectra.

Fig. 3 The ¹³C NMR spectrum of 5a

Fig. 4 The ¹³C NMR spectrum of 5e

Reviewer #2

Sure, impure CO₂ directly from flue gas or other sources has great practical application due to avoidance of the associated cost and energy requirement for the CO₂ purification procedure and provides a new direction for the application of the Carbon Capture and Utilization (CCU) principle. However, it also faces challenging, because the presence of impurities such as O₂, water vapor, CO, NO_x, and hydrocarbons in the CO₂ stream can be extremely harmful for the catalyst/reactants/intermediates and can be completely detrimental for the product formation. In this context, designing the reactions and catalyst systems would be crucial. **Please add comments and discussion on how to deal with these issues to circumvent impurity problems in this work.**

Response: We are grateful for the referee's recognition of this strategy. Initially, the reaction was conducted employing pure CO₂, yielding the desired product with an isolated yield of 90%. Upon transitioning to the utilization of exhaust gas, a noteworthy outcome emerged as we observed the formation of the γ -lactam compound as well, albeit with a minor reduction in yield (85%). As widely recognized, the [Ir(ppy)₂(dtbbpy)]PF₆ catalyst holds a well-established reputation within the realm of photocatalysis. Consequently, we wish to say our work extends and explores the compatibility boundaries of this catalyst, [Ir(ppy)₂(dtbbpy)]PF₆. We agree with the referee that the impurities could be very challenging in organic synthesis, however, in our case we did not observe any issue in this respect.

Additional discussion at the beginning of the Introduction on carbon capture and utilisation (CCU) (integration of CO₂ capture and subsequent conversion) to organic compounds such as HCCOH, MeOH and organic molecules can be traced back to 2011 (e.g. CO₂ capture and activation and its subsequent conversion, *Energy Environ. Sci.*, 2011, 4, 3971; *Angew. Chem. Int. Ed.*, 2012, 51, 11306), and recently published account (see, *Green Chem.*, 2016, 18, 5831; *Acc. Chem. Res.*, 2019, 52, 2892-2903, and references are therein) is also suggested.

Response: Thanks for this suggestion. As the reviewer suggested, we have checked all the mentioned articles and added them in the reference.

On the other hand, identification or trapping the radical species C, γ -amino benzylic anion species D is suggested to be done further, by using EPR or in situ spectroscopic techniques such as operando DRIFTS, or DFT study.

Response: The formation of product **3ac** (Fig. 2 in the manuscript) serves as compelling evidence for the presence of the benzylic radical species. Without the formation of this benzylic radical, the formation of the seven-member ring lactam product would be impossible. The initial participation of the first 1,1-diphenylethene acts as the receptor for the α -amino radical, thereby engendering the generation of the benzylic radical species (C). Subsequently, the capture of this radical by the second 1,1-diphenylethene molecule. Then, followed by a sequence of steps encompassing the reduction of benzylic radical to benzylic anion by Ir(II), carboxylation, and cyclization, culminating in the formation of the seven-member ring product (Fig. 5). We must mention that we tried the EPR experiments to trap this radical, however, we were not successful.

Fig. 5 The formation of **3ac**.

The existence of the γ -amino benzylic anion species *D* has been substantiated through comprehensive mechanism study experiments. These include “isotope labeling experiments with D_2O ”, “intercepting the carbanion intermediate with aldehyde”, and “Defluorinative alkylation of α -(Trifluoromethyl)-styrene” as showed in the mechanism studies part (**Fig. 4 Mechanistic studies**. In the manuscript). The outcomes of these experiments converge to furnish solid evidence for the formation of species *D*. To make the results clear to compare, I copied the results and shown below (**Fig. 6**):

Fig. 6 The copied Fig. 4 in manuscript

Reviewers' Comments:

Reviewer #1:

Remarks to the Author:

Thank you for the revision and comments. I mostly consented to and was satisfied with what the authors did to improve the clarity of the manuscript. One additional concern: for replying to comment 1), the authors made the table, in which it is also clarified that the reduction potential ($E(\text{re})$) of $[\text{Ir}(\text{ppy})_2(\text{dtbbpy})]^*$ is +0.66, which cannot oxidize ArNHR ($E(\text{ox}) = +0.81$). This suggests that $[\text{ArNR}]^-$, rather than neutral ArNHR , was indeed oxidized under basic conditions. If that is the case, the authors should slightly modify species in the catalytic cycle (Fig. 5). It is now convincing to read the authors' answer as to why basic conditions (different amounts of KHCO_3 : 0.2 eq to 3 eq, both worked similarly effectively) are fluctuant and have a lower limit but barely an upper limit

Thank you for the revision and comments. I mostly consented to and was satisfied with what the authors did to improve the clarity of the manuscript. One additional concern: for replying to comment 1), the authors made the table, in which it is also clarified that the reduction potential ($E(\text{re})$) of $[\text{Ir}(\text{ppy})_2(\text{dtbbpy})]^*$ is +0.66, which cannot oxidize ArNHR ($E(\text{ox}) = +0.81$). This suggests that $[\text{ArNR}]^-$, rather than neutral ArNHR, was indeed oxidized under basic conditions. If that is the case, the authors should slightly modify species in the catalytic cycle (Fig. 5). It is now convincing to read the authors' answer as to why basic conditions (different amounts of KHCO_3 : 0.2 eq to 3 eq, both worked similarly effectively) are fluctuant and have a lower limit but barely an upper limit.

Response: We appreciate the reviewer for this additional insight. We indeed considered the initial oxidation step to involve ArNHR rather than $[\text{ArNR}]^-$. This assertion was substantiated by our Stern-Volmer fluorescence quenching experiments, which clearly demonstrated the substantial quenching of the excited photocatalyst ($[\text{Ir}(\text{ppy})_2(\text{dtbbpy})\text{PF}_6]$) by varying concentrations of neutral ArNHR. This evidence strongly supported the oxidation of neutral ArNHR. To facilitate a clearer understanding of our findings, we have included **Figure 6** (below) in the Supplementary Information. Additionally, the reduction potential ($E(\text{re})$) of $[\text{Ir}(\text{ppy})_2(\text{dtbbpy})]^*$ is +0.66, and oxidation potential of ($E(\text{ox})$) ArNHMe is +0.81. The difference is quite small, so the value might fluctuate slightly under the experimental conditions. Therefore, we insist on keeping the original catalytic cycle (**Figure 5**).

Figure 6. Luminescence quenching experiments